# New Treatment for Type 2 Diabetes Mellitus Using a Novel Bipyrazole Compound

**DOI:** 10.3390/cells12020267

**Published:** 2023-01-09

**Authors:** Abdelrahim Alqudah, Esam Y. Qnais, Mohammed A. Wedyan, Sara Altaber, Rawan Abudalo, Omar Gammoh, Hakam Alkhateeb, Sajeda Bataineh, Rabaa Y. Athamneh, Muna Oqal, Kayed Abu-Safieh, Lana McClements

**Affiliations:** 1Department of Clinical Pharmacy and Pharmacy Practice, Faculty of Pharmaceutical Sciences, The Hashemite University, Zarqa 13133, Jordan; 2Department of Biology and Biotechnology, Faculty of Science, The Hashemite University, Zarqa 13133, Jordan; 3Department of Clinical Pharmacy and Pharmacy Practice, Faculty of Pharmacy, Yarmouk University, Irbid 21163, Jordan; 4Department of Basic Medical Sciences, Faculty of Medicine, Yarmouk University, Irbid 21163, Jordan; 5Department of Medical Laboratory Sciences, Faculty of Applied Health Sciences, The Hashemite University, Zarqa 13133, Jordan; 6Department of Medical Laboratory Sciences, Faculty of Allied Medical Sciences, Zarqa University, Zarqa 13133, Jordan; 7Department of Pharmaceutical Technology, Faculty of Pharmaceutical Sciences, The Hashemite University, Zarqa 13133, Jordan; 8Department of Chemistry, Faculty of Science, The Hashemite University, Zarqa 13133, Jordan; 9School of Life Sciences, Faculty of Science, University of Technology Sydney, Sydney, NSW 2007, Australia

**Keywords:** 2′,3,3,5′-tetramethyl-4′-nitro-2′H-1,3′-bipyrazole, insulin resistance, diabetes, oxidative stress, inflammation

## Abstract

2′,3,3,5′-Tetramethyl-4′-nitro-2′H-1,3′-bipyrazole (TMNB) is a novel bipyrazole compound with unknown therapeutic potential in diabetes mellitus. This study aims to investigate the anti-diabetic effects of TMNB in a high-fat diet and streptozotocin-(HFD/STZ)-induced rat model of type 2 diabetes mellitus (T2D). Rats were fed HFD, followed by a single low dose of STZ (40 mg/kg). HFD/STZ diabetic rats were treated orally with TMNB (10 mg/kg) or (200 mg/kg) metformin for 10 days before terminating the experiment and collecting plasma, soleus muscle, adipose tissue, and liver for further downstream analysis. TMNB reduced the elevated levels of serum glucose in diabetic rats compared to the vehicle control group (*p <* 0.001). TMNB abrogated the increase in serum insulin in the treated diabetic group compared to the vehicle control rats (*p <* 0.001). The homeostasis model assessment of insulin resistance (HOMA-IR) was decreased in the diabetic rats treated with TMNB compared to the vehicle controls. The skeletal muscle and adipose tissue protein contents of GLUT4 and AMPK were upregulated following treatment with TMNB (*p <* 0.001, < 0.01, respectively). TMNB was able to upregulate GLUT2 and AMPK protein expression in liver (*p* < 0.001, < 0.001, respectively). LDL, triglyceride, and cholesterol were reduced in diabetic rats treated with TMNB compared to the vehicle controls (*p* < 0.001, 0.01, respectively). TMNB reduced MDA and IL-6 levels (*p* < 0.001), and increased GSH level (*p* < 0.05) in diabetic rats compared to the vehicle controls. Conclusion: TMNB ameliorates insulin resistance, oxidative stress, and inflammation in a T2D model. TMNB could represent a promising therapeutic agent to treat T2D.

## 1. Introduction

Diabetes mellitus is a metabolic disorder, characterized by hyperglycemia, affecting carbohydrate, fat, and protein metabolism, resulting from abnormal insulin secretion, insulin resistance, or both of these factors [1]. The most frequent form of diabetes mellitus is type 2 diabetes mellitus (T2D), that encompasses over 80% of all people with diabetes mellitus. Compared to type 1 diabetes mellitus (T1D), where is a minute amount or no insulin remaining. T2D is characterized by insulin resistance and some insulin deficiency. It generally occurs in individuals during their adulthood, particularly those who are overweight or obese [2].

Insulin-mediated glucose uptake mainly occurs in skeletal muscles and adipose tissue, and this uptake is mediated by insulin-sensitive glucose transporter-4 (GLUT4) [3]. There is a large body of evidence indicating that the activation of AMP-activated protein kinase (AMPK) stimulates the translocation of GLUT4 to the cell surface, therefore facilitating glucose uptake by the skeletal muscle and adipose tissue via an insulin-independent mechanism [4]. Insulin resistance occurs when insulin is unable to facilitate the glucose uptake into skeletal muscle and adipose tissues through this translocation of GLUT4 to the cell membrane. Moreover, glucose transporter isoform 2 (GLUT2) is a key member of glucose transporter family in hepatocytes. In hepatocytes, GLUT2 controls the vast majority of glucose uptake relative to glucose concentration in the circulation, facilitating the convertsion of glucose to glycogen [5]. Additionally, the activation of AMPK in liver tissues leads to decreased hepatic glucose production [6]. Thus, the activation of the AMPK-GLUT4 pathway is an effective approach to treat T2D by enhancing insulin sensitivity [7].

It is well-known that T2D is considered an inflammatory condition, and inflammation and oxidative stress have been implicated in T2D pathogenesis and associated complications [8]. Pro-inflammatory cytokines including IL-6 are often elevated in obese people, and this increase is correlated with the development of insulin resistance and predictive of fully-established T2D [9,10]. Moreover, inflammation and oxidative stress can cause damage and death of pancreatic beta cells, which present in both T1D and T2D [11]. Additionally, diabetic complications including cardiovascular diseases are strongly correlated with increased expression of both inflammatory mediators and oxidative stress [12]. This suggests that inflammation and oxidative stress play a vital role in T2D development and its complications. 

Azoles are heterocyclic compounds with five-membered monocyclic heteroarenes containing one nitrogen atom and at least one other non-carbon atom, including nitrogen, oxygen, or sulfur, as a part of the ring structure including pyrazoles. These compounds are important nitrogen-heterocycles with pharmaceutical and agrochemical applications [13]. In 1883, the first pyrazole was generated through a well-known Knorr pyrazole synthesis reaction and demonstrated antipyretic bioactivity [14]. Pyrazole synthesis includes the condensation of a 1,3-dicarbonyl compound with hydrazine [15]. Several pharmacological agents contain pyrazole moiety including antimicrobials, hypoglycemic, and anti-inflammatory agents, due to their proven biological properties that lead to analgesic, anti-inflammatory, and anti-depressant effect [16]. Among pyrazoles, bipyrazoles have also shown beneficial therapeutic effects for treatment of infectious diseases, inflammatory conditions and some cancers [17]. 

2′,3,3,5′-Tetramethyl-4′-nitro-2′H-1,3′-bipyrazole (TMNB) is a new bipyrazole compound synthesized by classical Knorr pyrazole synthesis through the reaction between (0.17 g, 1 mmol) of 5-hydrazino-1,3-dimethyl-4-nitro-1H-pyrazole with (0.11 g, 1.1 mmol) of acetyl –acetone. The structure of this compound (Figure 1) was determined using infrared (IR) spectroscopy, HNMR, CNMR, and mass spectra [18]. The biological effect, particularly the anticancer activity, of this compound was only tested on human MCF-7 breast cancer cells and human K562 chronic myelogenous leukemia cells [18]. However, the anti-diabetic, anti-lipidemic, and anti-inflammatory effects of TMNB have not been studied yet. Therefore, the aim of this study was to investigate these effects of TMNB in a T2D rat model compared to vehicle treated controls and the gold standard hypoglycemic agent, metformin.

## 2. Materials and Methods

### 2.1. Lipinski’s Rule of Five for Drug-Likeness

Lipinski’s rule of five (Ro5) is a rule of thumb to evaluate the drug-likeness and to determine if a chemical compound with certain pharmacological or biological activity has properties that would make it a likely orally active drug in humans [19]. Ro5 depends on simple physiochemical property ranges: (i) the molecular weight, which should be less than 500 g/mol, (ii) lipophilicity (Log P) of less than 5, and (iii) a number of hydrogen bond donors and acceptors of less than 5 and 10, respectively. These parameters are important for intestinal permeability and aqueous solubility and determine the initial oral bioavailability. If a compound fails to fulfill the criteria for parameters of Ro5, then it is highly probable that it won’t be suitable for oral administration [20]. TMNB drug-likeness was tested using Webserver-Aided Drug Design By Artificial Intelligence And Classical Algorithm website (WADDAICA) [21].

### 2.2. Acute Oral Toxicity Test 

OECD/OCDE Test guidelines on Acute Oral Toxicity under a computer-guided Statistical Programme-AOT425statPgm, version 1.0 [22] were used to conduct the limit dose testing by assessing the up and down increment of 20 mg/kg for TMNB. A total of 5 female rats were selected out of 30 rats by systematic randomization techniques. The weight differences between the selected rats did not exceed ±10% of the mean initial weight of all the rats included in the experiment. The rats fasted overnight before a treatment dose was administered. At a time, one rat was selected and 20 mg/kg TMNB dissolved in 0.1% dimethyl sulfoxide (DMSO) was administered. A gastric feeding tube was used for the drug administration. Signs of regurgitation were observed for the first 5 min after administering each dose; once this was settled, the rats were kept in metabolic cages (one per cage). In the first 4 h after dosing, each rat was monitored closely every 15 min, then every 30 min for the consecutive 6 h, and daily for the consecutive 38 h for short-term outcomes, and the full 12 days for the long-term outcomes, including death. Changes in behavior due to acute oral toxicity were also observed and recorded for each rat.

### 2.3. Induction of T2D and Experimental Design

Animal experimental procedures were approved by the animal ethics committee at the Hashemite University (IRB number: 14/4/2021/2022, 14 April 2022) and were in accordance with the guidelines of the U.S. National Institutes of Health on the use and care of laboratory animals and with the Animal Research: Reporting of In Vivo Experiments (ARRIVE) guidelines (https://arriveguid elines.org, accessed 2 May 2022).

Thirty-week-old male adult Sprague-Dawley rats were maintained under standard conditions, including 12-hour light/dark cycles and a 22 ± 2° temperature [23]. T2D was induced by feeding the experimental rats HFD (60% fat) for 3 weeks, followed by one intraperitoneal injection of streptozotocin (STZ; 40 mg/kg) [24]. One week after STZ injection, plasma glucose was measured and rats with a plasma glucose concentration over 200 mg/dL were considered to have developed T2D and were selected for the subsequent experiments. 

Rats were randomly divided into four groups (*n* = 6 each) as follows: (i) normal non-diabetic control group (non-diabetic, ND) receiving normal diet, (ii) vehicle control (VC) diabetic group treated with dimethyl sulfoxide (DMSO, Panreac Quimica SA, Barcelona, Spain) only, (iii) diabetic group treated with 10 mg/kg 2′,3,3,5′-tetramethyl-4′-nitro-2′H-1,3′-bipyrazole (TMNB), and (iv) diabetic group treated with 200 mg/kg metformin (MeRCK, Darmstadt, Germany). All treatments were given orally once per day. After 10 days of treatment, rats were euthanized as per local standard operating procedures using carbon dioxide before blood and skeletal muscle (soleus muscle), adipose tissue, and liver were collected for ex-vivo analysis.

### 2.4. Biochemical Investigations

#### 2.4.1. Measurement of Serum Glucose, Insulin, and Lipids in Rat Serum

Serum glucose and insulin were determined using commercial kits glucose assay kit (Mybiosource, San Diego, CA, USA) and rat insulin ELISA kit (Mybiosource, San Diego, CA, USA), respectively, as per the manufacturer’s instructions. Triglyceride (TG, triglyceride assay kit), low density lipoproteins (LDL, LDL assay kit), and cholesterol (Total Cholesterol assay kit) were also measured using commercially available kits (Mybiosource, San Diego, CA, USA) according to the manufacturer’s instructions.

#### 2.4.2. Homeostasis Model Assessment of Insulin Resistance (HOMA-IR)

This model represents the interaction between fasting plasma insulin and fasting plasma glucose that is a useful tool for determining insulin resistance. In the current study, we used the following formula to compute HOMA-IR [25]:HOMA−IR=(Fasting glucose (mg/dl)×Fasting insulin (μIU/ml)/405 

#### 2.4.3. Measurement of Serum Glutathione (GSH), Malondialdehyde (MDA), and IL-6 Serum Concentrations

Reduced glutathione (GSH, GSH assay kit) and IL-6 (IL-6 ELISA kit) levels were measured in the serum using commercially available kits (Mybiosource, San Diego, CA, USA). Plasma MDA level was determined by using commercially available thiobarbituric acid (TBA) Assay Kit (Mybiosource, San Diego, CA, USA) according to the manufacturer’s instructions.

### 2.5. Western Blotting

Skeletal muscle tissues (soleus muscle), adipose tissue, and liver were homogenized in Radioimmunoprecipitation (RIPA)-lysis buffer, containing a protease inhibitor cocktail (Santa Cruz Biotechnology, Dallas, TX, USA), using a tissue homogenizer. Homogenates were centrifuged at 13,000 rpm for 20 min at 4 °C and supernatant was collected. Total protein was quantified using bicinchoninic acid assay (Bioquochem, Austurias, Spain). Equal amounts of protein was separated by sodium dodecyl sulfate-polyacrylamide gel then transferred onto nitrocellulose membrane (Thermo Fisher Scientific, Carlsbad, CA, USA). The membrane was blocked for 1 h at room temperature using 3% bovine serum albumin (BSA) before incubating overnight with either pAMPK-α1 (Abcam, Cambridge, UK), GLUT2, or GLUT4 (Mybiosource, San Diego, CA, USA) primary antibodies (1:1000 dilution). The membrane was washed three time with a washing buffer (Tween-20/Tris-buffered saline) before incubating it with the goat-anti-rabbit secondary antibody (Mybiosource, San Diego, CA, USA, 1:5000 dilution) for 1 h at room temperature. Following incubation, the membrane was washed three times before submerging it into the ECL substrate (ThermoScientific, Carlsbad, CA, USA) for one minute, followed by imaging with chemiLITE Chemiluminescence Imaging System (Cleaverscientific, Rugby, UK). To ensure equal protein gel loading, β-actin was used as a housekeeping gene (Mybiosource, San Diego, CA, USA, 1:10,000 dilution). The intensity of the bands was measured using Image J software and adjusted to β-actin. 

### 2.6. Statistical Analysis

All analyzed parameters were tested for normality of the data using Kolmogorov-Smirnov test. Data are represented as mean±SEM. Differences between groups were calculated using one-way analysis of variance (ANOVA) followed by Tukey post hoc test using Graphpad Prism software version (9.3.1). The significance value of difference was considered when the *p* value was less than 0.05.

## 3. Results

### 3.1. Lipinski’s Rule and Drug-Likeness

According to the results obtained using WADDAICA, the properties of TMNB are suitable for oral administration and are as follows: lipophilicity (log *p* value): 1.922, hydrogen bond acceptors: 4, hydrogen bond doners: 0, molecular mass: 235.106924656, and rotatable bond numbers are 2. Therefore, TMNB passes the Lipinski’s rule of five. 

### 3.2. Acute Toxicity Study

No deaths were recorded after the administration of 20 mg/kg of TMNB, based on both the short and long-term outcomes of the dose tolerance testing. Nevertheless, some behavioral signs of toxicity were observed, including tachypnoea, irritation, and restlessness. The lethal dose-50 (LD_50_) was calculated to be greater than 20 mg/kg using oral route (Table 1). 

### 3.3. The Hypoglycemic Effect of 2′,3,3,5′-Tetramethyl-4′-Nitro-2′H-1,3′-Bipyrazole (TMNB)

Serum glucose was significantly higher in the vehicle control diabetic group compared to the non-diabetic group (Figure 2A, *n* = 6, *p* < 0.001). Treatment with TMNB significantly reduced serum glucose concentrations compared to the vehicle controls in the presence of T2D (Figure 2A, *n* = 6, *p* < 0.001). Similarly, mean glucose concentration in the diabetic group was significantly reduced with metformin treatment compared to the vehicle controls (Figure 2A, *n* = 6, *p* < 0.01). Insulin levels were significantly increased in the vehicle control diabetic group compared to the non-diabetic group (Figure 2B, *n* = 6, *p* < 0.001); however, treating diabetic rats with TMNB significantly reduced the insulin compared to the vehicle control group (Figure 2B, *n* = 6, *p* < 0.001). Similar to TMNB, treating diabetic rats with metformin significantly reduced insulin levels compared to the vehicle control group (Figure 2B, *n* = 6, *p* < 0.001). No difference was observed between TMNB and metformin in terms of glucose and insulin systemic concentration.

In order to determine the effects of TMNB on insulin resistance, HOMA-IR was measured. The presence of T2D was confirmed by HOMA-IR, which significantly increased in the vehicle control diabetic group compared to the non-diabetic group (Figure 2C, *n* = 6, *p* < 0.001). Interestingly, TMNB was able to restore HOMA-IR in the diabetic group, which was comparable to the non-diabetic group. The same effect on HOMA-IR was observed when diabetic rats were treated with metformin. HOMA-IR was not different between the TMNB and metformin groups. 

To determine the mechanism by which TMNB improves blood glucose and insulin resistance, GLUT4 and AMPK protein expression were measured in skeletal muscle tissue. GLUT4 protein expression was significantly downregulated in the presence of T2D (Figure 3A, *n* = 6, *p* < 0.001); however, treating diabetic rats with TMNB significantly upregulated GLUT4 expression compared to the vehicle control diabetic group (Figure 3A, *n* = 6, *p* < 0.001). Metformin was also able to upregulate GLUT4 expression compared to the vehicle control diabetic group (Figure 3A, *n* = 6, *p* < 0.001). Similarly, AMPK expression was significantly downregulated as a result of T2D (Figure 3B, *n* = 6, *p* < 0.001), which was abrogated with TMNB or metformin (Figure 3B, *n* = 6, *p* < 0.001). No difference was observed in GLUT4 and AMPK expression between TMNB and metformin groups.

Moreover, GLUT4 and AMPK protein expression in adipose tissue were measured. GLUT4 protein expression was significantly downregulated in the presence of T2D (Figure 4A, *n* = 6, *p* < 0.05); whereas administration of TMNB significantly upregulated GLUT4 expression compared to the vehicle controls (Figure 4A, *n* = 6, *p* < 0.01). Metformin was also able to upregulate GLUT4 expression compared to the vehicle control diabetic group (Figure 4A, *n* = 6, *p* < 0.001). Similarly, AMPK expression was significantly downregulated as a result of T2D (Figure 4B, *n* = 6, *p* < 0.001), and treating diabetic rats with either TMNB or metformin significantly upregulated AMPK expression compared to the vehicle control diabetic group (Figure 4B, *n* = 6, *p* < 0.01, < 0.05, respectively). No difference was observed in GLUT4 and AMPK expression between TMNB and metformin groups.

To further elucidate the mechanism by which TMNB improves insulin sensitivity and reduces glucose levels, hepatic glucose transporter 2 (GLUT2) and AMPK protein expression were measured. GLUT2 protein expression was significantly downregulated in the presence of T2D (Figure 5A, *n* = 6, *p* < 0.001); whereas, TMNB significantly upregulated GLUT2 expression compared to the vehicle control diabetic group (Figure 5A, *n* = 6, *p* < 0.01). Metformin was also able to upregulate GLUT2 expression compared to vehicle controls (Figure 5A, *n* = 6, *p* < 0.01). Similarly, AMPK expression was significantly downregulated as a result of T2D (Figure 5B, *n* = 6, *p* < 0.001), and treating diabetic rats with either TMNB or metformin significantly upregulated AMPK expression compared to vehicle controls (Figure 5B, *n* = 6, *p* < 0.001). No difference was observed in GLUT2 and AMPK expression between TMNB and metformin groups.

### 3.4. The Effect of 2′,3,3,5′-Tetramethyl-4′-Nitro-2′H-1,3′-Bipyrazole (TMNB) on Lipid Profile

As depicted in Figure 6, dyslipidemia was clearly present in diabetic rats. LDL (Figure 6A, *n* = 6, *p* < 0.001), total cholesterol (Figure 6B, *n* = 6, *p* < 0.05), and triglycerides (TGs; Figure 6C, *n* = 6, *p* < 0.01) were significantly higher in the vehicle control diabetic group compared to the non-diabetic group. Treating diabetic rats with TMNB significantly reduced serum LDL (Figure 6A, *n* = 6, *p* < 0.001), total cholesterol (Figure 6B, *n* = 6, *p* < 0.01), and TGs (Figure 6C, *n* = 6, *p* < 0.001systemic concentration compared to the vehicle control diabetic group. Similarly, metformin was able to significantly reduce LDL (Figure 6A, *n* = 6, *p* < 0.05), cholesterol (Figure 6B, *n* = 6, *p* < 0.01), and TGs (Figure 6C, *n* = 6, *p* < 0.001) levels in diabetic rats compared to vehicle controls. 

### 3.5. The Effect of 2′,3,3,5′-Tetramethyl-4′-Nitro-2′H-1,3′-Bipyrazole (TMNB) on MDA, GSH, and IL-6

Serum MDA (Figure 7A, *n* = 6, *p* < 0.001) and IL-6 (Figure 7B, *n* = 6, *p* < 0.001) concentrations were significantly increased in the presence of T2D. Interestingly, TMNB showed an ability to significantly reduce serum MDA and IL-6 levels in diabetic rats compared to the vehicle control group (Figure 7A,B, *n* = 6, *p* < 0.001). The same effect was observed when diabetic rats were treated with metformin (Figure 7A,B, *n* = 6, *p* < 0.001). On the other hand, serum GSH expression was significantly reduced in vehicle control diabetic rats compared to non-diabetic rats (Figure 7C, *n* = 6, *p* < 0.001); whereas, treating diabetic rats with either TMNB or metformin demonstrated a significant increase in GSH compared to the vehicle control diabetic group (Figure 7C, *n* = 6, *p* < 0.05, 0.01, respectively).

## 4. Discussion

Our study investigated the role of TMNB in improving glucose and lipid profile in HDF/STZ-induced T2D rats. High blood glucose and insulin resistance are the main features of T2D [26], and the hyperglycemia is a major cause of diabetes complications, including cardiovascular diseases, nephropathy, retinopathy, and neuropathy [27]. In our study, the results demonstrated that TMNB significantly reduced systemic glucose concentration, which was comparable to the gold standard drug, metformin, suggesting that TMNB represents a promising hypoglycemic candidate. Moreover, our results report an increase in systemic insulin concentration in the vehicle control diabetic group. Thus, the HFD/STZ T2D rat model resembles the characteristics of T2D including early-stage hyperinsulinemia, hyperglycemia, hyperlipidaemia, and beta cells dysfunction [28]. Several studies reported that hyperinsulinemia exists in T2D because the peripheral tissues lack their insulin sensitizing properties. This ultimately results in hyperglycemia; and an increase in insulin secretion in the early stages of T2D as part of the compensatory mechanism that aims to counteract the presence of insulin resistance in T2D [29,30,31,32]. On the other hand, some reports showed that dimethyl sulfoxide (DMSO) could also increase insulin secretion. Kemp and Habener, 2002, demonstrated that 0.5–2.5% DMSO significantly enhanced insulin secretion and insulin gene expression in the INS beta cell line, which could be through direct molecular interaction with downstream target of the glucagon-like peptide 1 (GLP-1) receptor signalling pathway in the beta cell [33]. A recent study reported that enhanced insulin secretion was observed in a mouse insulinoma model of pancreatic beta cell, MIN6-K8, when exposed to high glucose medium in association with enhanced calcium influx, suggesting that DMSO’s mechanism of action is upstream of calcium-dependant insulin granule exocytosis [34]. These results also suggest that the observed increase in insulin levels in the vehicle control group in our study might be the effect of DMSO injection. Interestingly, TMNB was potent at reducing serum insulin to normal/non-diabetic levels. The same effect was observed when diabetic rats were treated with the gold standard drug, metformin. TMNB was also able to abrogate insulin resistance, measured by HOMA-IR, which showed that TMNB improved insulin sensitivity in diabetic rats, and this is in agreement with the reduction in serum insulin observed after treatment with TMNB.

Skeletal muscle and adipose tissue are responsible for 50–80% of glucose transportation [35,36]. GLUT-4 is a critical glucose transporter facilitating utilization of extracellular glucose by insulin-sensitive cells, hence maintaining blood glucose homeostasis [37]. Moreover, reduced GLUT-4 skeletal muscle and adipose protein expression and inhibition of GLUT4 translocation to the cell surface results in insufficient glucose transportation, and hence, insulin resistance. The activation of the AMPK-GLUT4 pathway enhances insulin sensitivity and it has been shown to improve glucose control in T2D [38,39]. In addition, the role of AMPK in the prevention of T2D has previously been investigated in combination with the regulation of insulin signaling and GLUT-4 activity [40]. Our results demonstrate that TMNB is capable of increasing the expression of both AMPK and GLUT4 in skeletal muscles and adipose tissue, suggesting that TMNB can improve glucose uptake through the AMPK-GLUT4 pathway. Moreover, GLUT2 is responsible for bulk glucose uptake in the liver and is required for the physiological control of glucose-sensitive genes. Inactivation of GLUT2 in the liver leads to impaired glucose-stimulated insulin secretion [41,42]. Furthermore, AMPK activation in the liver reduces hepatic glucose production. Interestingly, TMNB is able to activate GLUT2 and AMPK in the liver, which further explains the hypoglycemic mechanism of TMNB.

Many pyrazole and bipyrazole compounds in the preclinical studies of T1D exerted their antioxidant, anti-inflammatory, and hypoglycemic effects by lowering the blood glucose level or by regulating the peroxisome proliferator-activated receptor (PPARγ), sodium glucose co-transporter (SGLT1), and inhibiting some of the advanced glycation end product (RAGE) [43]. In addition to hyperglycemia, T2D is associated with dyslipidemia. High postprandial TGs, total cholesterol, and LDL define diabetic dyslipidemia [44]. These lipid alterations are the key factors leading to T2D-associated complications [45]. Particularly, dyslipidemia is a significant risk factor for macrovascular diabetes complications, and numerous studies have linked dyslipidemia to microvascular complications associated with T2D, such as diabetic retinopathy, nephropathy, and neuropathy [44]. This study findings revealed that TMNB was able of reducing TGs, total cholesterol, and LDL in our T2D model, suggesting that TMNB may reduce cardiovascular disease risk associated with T2D. This should be explored in future studies.

Many clinical and experimental studies show that there is a strong link between oxidative stress and the development of T2D and its complications [46]. Oxidative stress is defined as a reduced tolerance between oxidants and antioxidants, due to the production of reactive oxygen species (ROS) and the reduction in the rate of antioxidant defense mechanism, including GSH (non-enzymatic antioxidant) [47]. ROS can damage the lipids, causing lipid peroxidation of low-density lipoprotein (oxLDL) or peroxidation of polyunsaturated fatty acid (oxPUFAs). Additionally, ROS induces the release of MDA, a highly reactive compound that interacts with proteins, nucleic acids, and causing damage to various tissues and cells [48]. MDA has been used as a biomarker of lipid peroxidation and as an indication of free radical damage in the blood [49]. Our findings show that TMNB considerably reduces plasma MDA levels and increases GSH levels, implying that TMNB could be effective as an antioxidant agent in T2D by reducing lipid peroxidation or by increasing free radical scavenging activity.

Subclinical chronic inflammation has been implicated as an independent risk factor for the development and progression of T2D and its complications [8]. In particular, the multifunctional cytokine, IL-6, has been linked to the pathogenesis of T2D. Increased levels of systemic IL-6 is a strong predictor of T2D and is shown to have a role in the development of inflammation, insulin resistance, and beta cell dysfunction [50]. In addition, mounting data shows that IL-6 impairs insulin signaling in hepatocytes and inhibits glucose-stimulated insulin release from the pancreatic beta cell [50]. Our findings indicate that TMNB significantly reduced IL-6 levels, implying that it has an anti-inflammatory effect, which could improve T2D outcomes.

Limitations of this study include the following aspects: (i) TMNB was administered for a short period of time, (ii) GLUT4 expression was assessed using immunoblotting reflective of its total amount; however, immunohistochemistry may be a better technique to assess its activity and translocation to the cell membrane, and (iii) insulin secretion studies in vitro (mouse/human cell line and/or primary islets) and in vivo (GSIS on treated mice) should be studied in the future. Nevertheless, our findings in this study indicate the crucial role of TMNB in improving typical features of T2D, which is the first report to date.

## 5. Conclusions

In conclusion, our results demonstrate that 2′,3,3,5′-Tetramethyl-4′-nitro-2′H-1,3′-bipyrazole could be a very useful hypoglycemic agent for the treatment of T2D due to its multifactorial effects, including (i) the reduction in insulin resistance, (ii) increase in glucose uptake by the skeletal muscle, (iii) improvement in the lipid profile, (iv) reduction in oxidative stress and inflammation, and (v) the activation of the GLUT4-AMPK pathway. The effects and mechanisms demonstrated by TMNB were very similar to the gold standard treatment, metformin.

## Figures and Tables

**Figure 1 cells-12-00267-f001:**
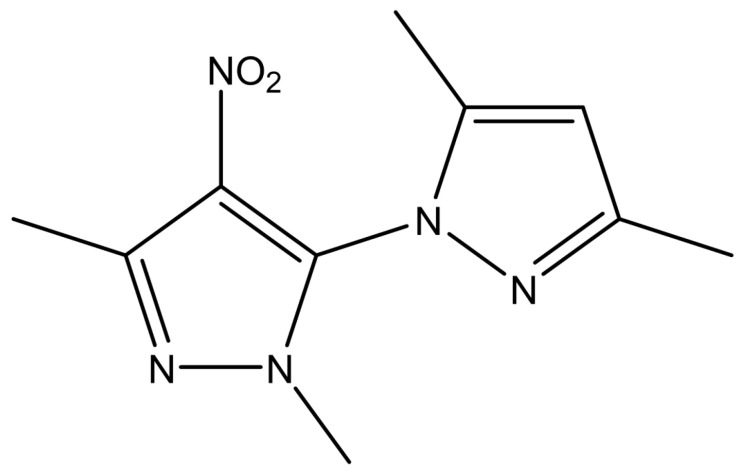
2′,3,3,5′-Tetramethyl-4′-nitro-2′H-1,3′-bipyrazole (TMNB) chemical structure.

**Figure 2 cells-12-00267-f002:**
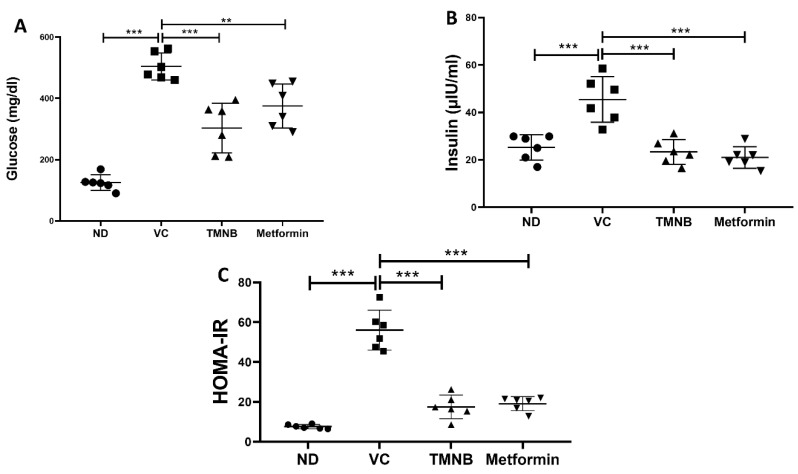
The anti-diabetic effect of 2′,3,3,5′-tetramethyl-4′-nitro-2′H-1,3′-bipyrazole. TMNB significantly reduced glucose (**A**) and insulin (**B**) levels in diabetic rats. HOMA-IR (**C**) was significantly reduced with TMNB treatment. Rats were fed HFD for 3 weeks, followed by a single dose of STZ injection (40 mg/kg); and once diabetes was confirmed, rats were treated with 10 mg/kg TMNB or 200 mg/kg metformin for 10 days. After the end of the experiment, serum was collected for ELISA analyses. One-way ANOVA was followed by Tukey post-hoc multiple comparison test, ** < 0.01, *** < 0.001. ND; non-diabetic, VC; vehicle control, TMNB; 2′,3,3,5′-Tetramethyl-4′-nitro-2′H-1,3′-bipyrazole.

**Figure 3 cells-12-00267-f003:**
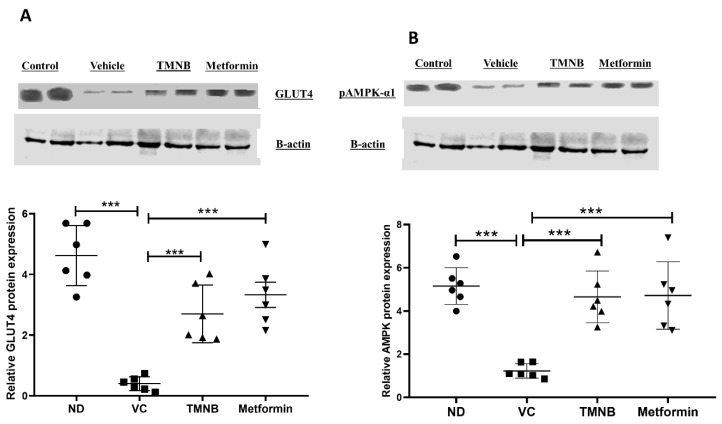
2′,3,3,5′-Tetramethyl-4′-nitro-2′H-1,3′-bipyrazole upregulated GLUT4 and AMPK expression in soleus muscle. TMNB significantly upregulated GLUT4 (**A**) and AMPK (**B**) expression within soleus muscle in diabetic rats. Rats were fed HFD for 3 weeks, followed by a single dose of STZ injection (40 mg/kg); once diabetes was confirmed, rats were treated with 10 mg/kg TMNB or 200 mg/kg metformin for 10 days. After the end of the experiment following euthanasia, soleus muscle was isolated and homogenized for downstream western blotting. One-way ANOVA was followed by Tukey post-hoc multiple comparison test, *** < 0.001. ND; non-diabetic, VC; vehicle control, TMNB; 2′,3,3,5′-Tetramethyl-4′-nitro-2′H-1,3′-bipyrazole.

**Figure 4 cells-12-00267-f004:**
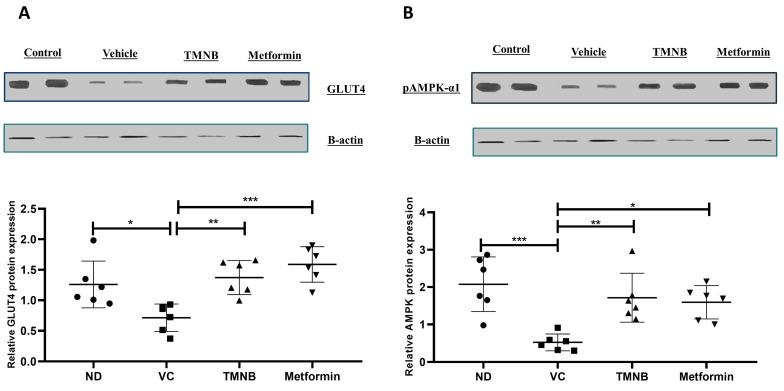
2′,3,3,5′-Tetramethyl-4′-nitro-2′H-1,3′-bipyrazole upregulated GLUT4 and AMPK expression in adipose tissue. TMNB significantly upregulated adipose tissue GLUT4 (**A**) and AMPK (**B**) expression in diabetic rats. Rats were fed HFD for 3 weeks, followed by a single dose of STZ injection (40 mg/kg); after diabetes was confirmed, rats were treated with 10 mg/kg TMNB or 200 mg/kg metformin for 10 days. At the end of the experiment following euthanasia, adipose tissue was isolated and homogenized before western blotting was performed. One-way ANOVA was followed by Tukey post-hoc multiple comparison test, * < 0.05, ** < 0.01, *** < 0.001. ND; non-diabetic, VC; vehicle control, TMNB; 2′,3,3,5′-Tetramethyl-4′-nitro-2′H-1,3′-bipyrazole.

**Figure 5 cells-12-00267-f005:**
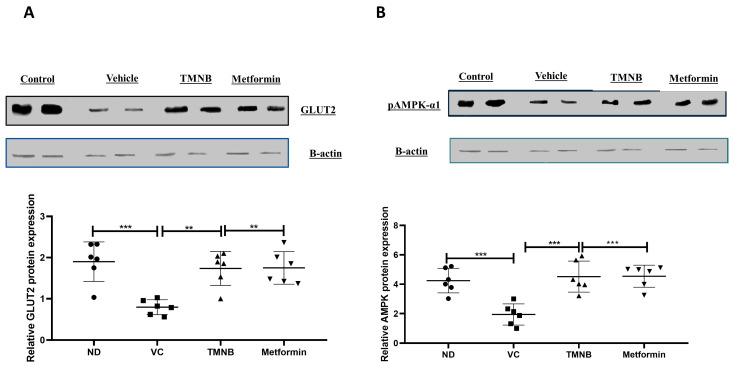
2′,3,3,5′-Tetramethyl-4′-nitro-2′H-1,3′-bipyrazole upregulated GLUT2 and AMPK expression in liver. TMNB significantly upregulated hepatic GLUT2 (**A**) and AMPK (**B**) expression in diabetic rats. Rats were fed HFD for 3 weeks, followed by a single dose of STZ injection (40 mg/kg); and once diabetes was confirmed, rats were treated with 10 mg/kg TMNB or 200 mg/kg metformin for 10 days. At the end of the experiment following euthanasia, liver was isolated and homogenized before being used western blotting. One-way ANOVA was followed by Tukey post-hoc multiple comparison test, ** < 0.01, *** < 0.001. ND; non-diabetic, VC; vehicle control, TMNB; 2′,3,3,5′-Tetramethyl-4′-nitro-2′H-1,3′-bipyrazole.

**Figure 6 cells-12-00267-f006:**
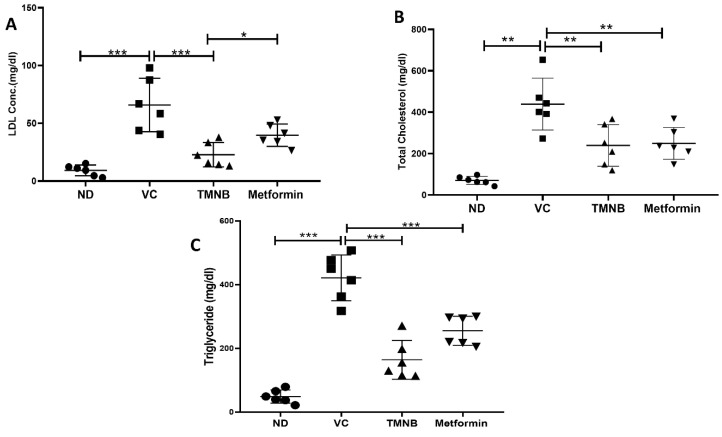
2′,3,3,5′-Tetramethyl-4′-nitro-2′H-1,3′-bipyrazole improves lipid profile in diabetes. TMNB significantly reduced LDL (**A**), cholesterol (**B**), and triglyceride (**C**) systemic concentrations in diabetic rats. Rats were fed HFD for 3 weeks, followed by a single dose of STZ injection (40 mg/kg); and once diabetes was confirmed, rats were treated with 10 mg/kg TMNB or 200 mg/kg metformin for 10 days. At the end of the experiment following euthanasia, serum was collected for ELISA analyses. One-way ANOVA was followed by Tukey post-hoc multiple comparison test, * < 0.05, ** < 0.01, *** < 0.001. ND; non-diabetic, VC; vehicle control, TMNB; 2′,3,3,5′-Tetramethyl-4′-nitro-2′H-1,3′-bipyrazole.

**Figure 7 cells-12-00267-f007:**
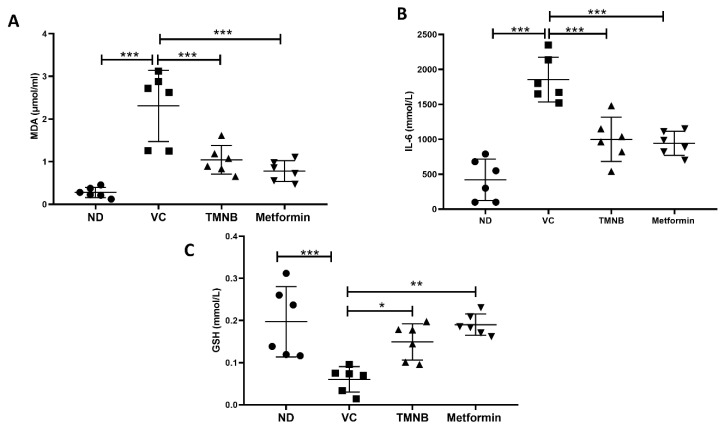
The anti-inflammatory effect of 2′,3,3,5′-Tetramethyl-4′-nitro-2′H-1,3′-bipyrazole. TMNB significantly reduced MDA (**A**) and IL-6 (**B**) levels and increased GSH (**C**) systemic concentration in diabetic rats. Rats were fed HFD for 3 weeks, followed by a single dose of STZ injection (40 mg/kg); and once diabetes was confirmed, rats were treated with 10 mg/kg TMNB or 200 mg/kg metformin for 10 days. After the end of the experiment following euthanasia, serum was collected for ELISA analysis. One-way ANOVA was followed by Tukey post-hoc multiple comparison test, * < 0.05, ** < 0.01, *** < 0.001. ND; non-diabetic, VC; vehicle control, TMNB; 2′,3,3,5′-Tetramethyl-4′-nitro-2′H-1,3′-bipyrazole.

**Table 1 cells-12-00267-t001:** Sequence and results of limit dose test of TMNB in rats.

Test Sequence	Animal ID	Dose (mg/kg)	Short-Term Result (48 h)	Long-Term Result (12 Days)
1	01	20	Survival	Survival
2	02	20	Survival	Survival
3	03	20	Survival	Survival
4	04	20	Survival	Survival
5	05	20	Survival	Survival

## Data Availability

The dataset generated during and/or analyzed during the current study are available from the corresponding author on reasonable request.

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
