# Peer review of "New Treatment for Type 2 Diabetes Mellitus Using a Novel Bipyrazole Compound"

_cells, 2023, doi:10.3390/cells12020267_

Round 1

Reviewer 1 Report

In this study, the authors focus on a novel bipyrazole compound (TMNB) that improves insulin resistance, increases the expression of GLUT4 and AMPK, and lowers LDL, total cholesterol, and triglycerides. They used an STZ/HFD-treated T2D rat model and performed all experiments on this single model. While the study tests a novel pharmacological agent and demonstrates beneficial effects comparable to the clinically used metformin, using only a single model and not testing other available tissue at least in that present model is interesting and limits the study. As the authors discuss in the limitations section additional experiments could easily be performed.

Comments from this Reviewer are below:

Comments:

1- The authors describe the insulin levels in the vehicle control group as unexpected but did not discuss it further. The difference between the non-diabetic (ND) and vehicle control (VC) groups is the diet (chow vs HFD), STZ treatment, and DMSO administration. Is the increase of insulin solely dependent on the development of insulin resistance? Is there any possibility that DMSO might cause an increase in insulin levels? There is literature discussing an acute increase in insulin following DMSO treatment. The authors should discuss this.

2- Metformin exerts its glucose-lowering effect mainly by inhibiting hepatic neogenesis.  As has been discussed in the limitations, studying the effects only in skeletal muscle and not in the liver, (and adipose tissue) is a real limitation, especially where the liver and adipose tissue was available for the authors. This reviewer assumes that those tissues were collected at the end of the study and checking GLUT4, AMPK (which metformin might act), and molecules for gluconeogenesis will provide more information for TNMB and improve the significance of the potential mechanism of action.

3- Did the authors check insulin secretion in vitro (pancreatic rat/mouse or human islets) or in vivo (rat/mouse-glucose stimulated insulin secretion-GSIS) in the presence of TNMB? This is a straightforward experiment that can give valuable information regarding the effects of TNMB on pancreatic islets.

Minor Comments

1- On line 213, correct “TMNNB”

2- Correct figure numbers between lines 203 and 213. All figure numbers should be “Figure 3” instead of Figure 2!

Author Response

Dear editor and reviewer

We would like to thank you for the invested time and effort in carefully reviewing our manuscript. We are grateful for giving us the opportunity to revise our manuscript. Your comments were very useful and helped us in improving our manuscript. After careful consideration of the comments, the revision included many positive changes as suggested.

Changes are indicated in the tracked changes mode.

In this study, the authors focus on a novel bipyrazole compound (TMNB) that improves insulin resistance, increases the expression of GLUT4 and AMPK, and lowers LDL, total cholesterol, and triglycerides. They used an STZ/HFD-treated T2D rat model and performed all experiments on this single model. While the study tests a novel pharmacological agent and demonstrates beneficial effects comparable to the clinically used metformin, using only a single model and not testing other available tissue at least in that present model is interesting and limits the study. As the authors discuss in the limitations section additional experiments could easily be performed.

Comments from this Reviewer are below:

Comments:

C1: The authors describe the insulin levels in the vehicle control group as unexpected but did not discuss it further. The difference between the non-diabetic (ND) and vehicle control (VC) groups is the diet (chow vs HFD), STZ treatment, and DMSO administration. Is the increase of insulin solely dependent on the development of insulin resistance? Is there any possibility that DMSO might cause an increase in insulin levels? There is literature discussing an acute increase in insulin following DMSO treatment. The authors should discuss this.

R1: Thank you for your comment and the insight about the effect of DMSO on insulin level. More details are added now to the discussion in the revised manuscript to clarify the increase in the insulin levels in vehicle control group including the possible effect of DMSO

C2: Metformin exerts its glucose-lowering effect mainly by inhibiting hepatic neogenesis.  As has been discussed in the limitations, studying the effects only in skeletal muscle and not in the liver, (and adipose tissue) is a real limitation, especially where the liver and adipose tissue was available for the authors. This reviewer assumes that those tissues were collected at the end of the study and checking GLUT4, AMPK (which metformin might act), and molecules for gluconeogenesis will provide more information for TNMB and improve the significance of the potential mechanism of action.

R2: The authors would like to thank you for this comment. As you stated that during the experiment, we collected adipose and liver tissues from the rats. And to address your valuable comment, western blotting was performed to assess the expression of AMPK and GLUT4 in adipose tissue and the results are presented now on Figure 4 in the revised manuscript. Also, because GLUT2 is responsible for glucose uptake in the liver, its expression was assessed in addition to the expression of AMPK in the liver which is now presented in Figure 5 in the revised manuscript. Interestingly, TNMP was able to upregulate GLUT4 and AMPK in adipose tissue and upregulate GLUT4 and AMPK in liver. The relevant details for the new results are added now to the introduction, methods, results and discussion in the revised manuscript.

C3: Did the authors check insulin secretion in vitro (pancreatic rat/mouse or human islets) or in vivo (rat/mouse-glucose stimulated insulin secretion-GSIS) in the presence of TNMB? This is a straightforward experiment that can give valuable information regarding the effects of TNMB on pancreatic islets.

R3: Thank you for your comment. We totally agree with you that this experiment is valuable and will strengthen our study, however, pancreatic rat/mouse or human islets are not available in our lab and we have limited fund to perform it at this stage, so, this experiment will be on our plan for future study and we added this to the limitations of our study.

Minor Comments

C1:  On line 213, correct “TMNNB”

R1: Corrected

C2:  Correct figure numbers between lines 203 and 213. All figure numbers should be “Figure 3” instead of Figure 2!

R2: Apology for this mistake. Its corrected now.

Reviewer 2 Report

Abstract

Use the journal style or format ie delete the words: Background, Material and methods, Results and conclusion.

Introduction

Line 83 the word (Figure 1) change to line 82  after " of this compound"

Lines 88, 126, 183,215,221,268, 342  change Tetramethyl- as tetramethyl

Line 127 change Metformin as metformin

Line 129 change CO2 as CO2

Line 139 change HOmeostasis  as Homeostasis

Line 148 change Thiobarbituric; Assay as thiobarbituric; assay

Line 180 change  LD50 as LD50

Author Response

Dear editor and reviewer

We would like to thank you for the invested time and effort in carefully reviewing our manuscript. We are grateful for giving us the opportunity to revise our manuscript. Your comments were very useful and helped us in improving our manuscript. After careful consideration of the comments, the revision included many positive changes as suggested.

Changes are indicated in the tracked changes mode.

Abstract

C1: Use the journal style or format ie delete the words: Background, Material and methods, Results and conclusion.

R1: Thank you for your comment. Its corrected now.

Introduction

C2: Line 83 the word (Figure 1) change to line 82  after " of this compound"

R2: Thanks for this comment. The word Figure 1 is moved after ‘’to this compound’’.

C3: Lines 88, 126, 183,215,221,268, 342  change Tetramethyl- as tetramethyl

R3: Apology for this mistake. All are corrected now.

C4: Line 127 change Metformin as metformin

R4: It is corrected now.

C5: Line 129 change CO2 as CO2

R5: It is corrected now.

C6: Line 139 change HOmeostasis  as Homeostasis

R6: Corrected now.

C7: Line 148 change Thiobarbituric; Assay as thiobarbituric; assay

R7: Corrected now.

C9: Line 180 change  LD50 as LD50

R9: Corrected now.

Reviewer 3 Report

In this study, a novel bipyrazole compound is discovered for the treatment of T2DM. I have two questions to be need responses:

1. Please see table 1, why did authors give a dose gradually to rats. For example, doses 10 mg/kg, 20 mg/kg and 30 mg/kg are used for their experiments. In this way, it can deduce the suitable toxic dose for the rat.

2. Please figure 1, it needs to know whether TMNB is a drug-like and Pan-assay interference compound (PAINS). For example, authors can calculate Lipinski's rule of five and PAINS on the webserver https://heisenberg.ucam.edu:5000/drugproperties 

Author Response

Dear editor and reviewer

We would like to thank you for the invested time and effort in carefully reviewing our manuscript. We are grateful for giving us the opportunity to revise our manuscript. Your comments were very useful and helped us in improving our manuscript. After careful consideration of the comments, the revision included many positive changes as suggested.

Changes are indicated in the tracked changes mode.

In this study, a novel bipyrazole compound is discovered for the treatment of T2DM. I have two questions to be need responses:

C1: Please see table 1, why did authors give a dose gradually to rats. For example, doses 10 mg/kg, 20 mg/kg and 30 mg/kg are used for their experiments. In this way, it can deduce the suitable toxic dose for the rat.

R1: Thank you for your comment. The authors totally agree with you that giving a gradual dose to rats will deduce the toxic dose, however, our idea behind using one dose is to minimize the number of animals required for the test, so, we decided to use 20 mg/kg dose for the toxicity test which is double the dose that is used for the treatment.

C2: Please figure 1, it needs to know whether TMNB is a drug-like and Pan-assay interference compound (PAINS). For example, authors can calculate Lipinski's rule of five and PAINS on the webserver https://heisenberg.ucam.edu:5000/drugproperties 

R2: The authors would like to thank you for this comment. The drug-likeness of TMNB was tested using the provided website and according to the results, it passes the Ro5. The relevant information is added now to the methods and results section in the revised manuscript.

Round 2

Reviewer 1 Report

Major comments 1, 2, and minor comments from this reviewer have been addressed. For comment 3, in the "limitations" section, would be great to add insulin secretion studies in vitro (mouse/human cell line and/or primary islets) and in vivo (GSIS on treated mice) as future plans.

Figure 4 and Figure 5 Legend titles include the tissue name (adipose tissue and liver). Please add the name of the tissue (Soleus muscle) yo the Figure 3 title.

Author Response

Dear editor and reviewer

We would like to thank you for the invested time and effort in carefully reviewing our manuscript. We are grateful for giving us the opportunity to revise our manuscript. Your comments were very useful and helped us in improving our manuscript. After careful consideration of the comments, the revision included many positive changes as suggested.

Changes are indicated in the tracked changes mode.

C1: Major comments 1, 2, and minor comments from this reviewer have been addressed. For comment 3, in the "limitations" section, would be great to add insulin secretion studies in vitro (mouse/human cell line and/or primary islets) and in vivo (GSIS on treated mice) as future plans.

R1: Many thanks for your comment. The suggested insulin secretion studies are added now to the limitations section as a future plan.

C2: Figure 4 and Figure 5 Legend titles include the tissue name (adipose tissue and liver). Please add the name of the tissue (Soleus muscle) yo the Figure 3 title.

R2: Thank you for your suggestion. The tissue name is added to the figures legends.

Reviewer 3 Report

The reference 21 for WADDAICA should be modified as follows:

Bai, Q.*, Ma, J., Liu, S., Xu, T., Banegas-Luna, A. J., Pérez-Sánchez, H.*, et al. WADDAICA: A webserver for aiding protein drug design by artificial intelligence and classical algorithm. Computational and Structural Biotechnology Journal 19, 3573-3579, (2021).  https://doi.org/10.1016/j.csbj.2021.06.017   

After above minor revision, I think this paper can be accepted.

Author Response

Dear editor and reviewer

We would like to thank you for the invested time and effort in carefully reviewing our manuscript. We are grateful for giving us the opportunity to revise our manuscript. Your comments were very useful and helped us in improving our manuscript. After careful consideration of the comments, the revision included many positive changes as suggested.

Changes are indicated in the tracked changes mode.

C1: The reference 21 for WADDAICA should be modified as follows:

Bai, Q.*, Ma, J., Liu, S., Xu, T., Banegas-Luna, A. J., Pérez-Sánchez, H.*, et al. WADDAICA: A webserver for aiding protein drug design by artificial intelligence and classical algorithm. Computational and Structural Biotechnology Journal 19, 3573-3579, (2021).  https://doi.org/10.1016/j.csbj.2021.06.017 

R1: Thank you for your comment. Reference number 21 is now modified as suggested.  

C: After above minor revision, I think this paper can be accepted.

R: Thank you very much.